# The Role of Oxidative Stress Markers in Predicting Acute Thrombotic Occlusion of Haemodialysis Vascular Access and Progressive Stenotic Dysfunction Demanding Angioplasty

**DOI:** 10.3390/antiox10040569

**Published:** 2021-04-08

**Authors:** Jenq-Shyong Chan, Po-Jen Hsiao, Wen-Fang Chiang, Prabir Roy-Chaudhury

**Affiliations:** 1Department of Internal Medicine, Division of Nephrology, Armed Forces Taoyuan General Hospital, Taoyuan 325, Taiwan; c120536077@aftygh.gov.tw (J.-S.C.); wfc96076@aftygh.gov.tw (W.-F.C.); 2Department of Medicine, Division of Nephrology, Tri-Service General Hospital, National Defense Medical Center, Taipei 114, Taiwan; 3National Defense Medical Center, School of Medicine, Taipei 114, Taiwan; 4Institute of Clinical Medicine, National Yang-Ming University, Taipei 112, Taiwan; 5Department of Life Sciences, National Central University, Taoyuan 320, Taiwan; 6Department of Medicine, Division of Nephrology and Hypertension, University of North Carolina, Chapel Hill, NC 27599, USA; 7Salisbury VA Medical Center, Salisbury, NC 27284, USA

**Keywords:** haemodialysis vascular access, vascular access dysfunction, oxidative stress markers, asymmetrical dimethylarginine (ADMA), peripheral arterial occlusive disease

## Abstract

Haemodialysis vascular access (VA) dysfunction is a major cause of morbidity in haemodialysis (HD) patients. Primary venous outflow occlusion and restenosis after percutaneous transluminal angioplasty (PTA) are two major obstacles for the long-term use of dialysis VA. It remains unclear whether oxidative stress markers can be used as predictors for thrombotic occlusion of VA and progressive stenosis dysfunction demanding PTA. All routine HD patients at one teaching hospital participated in this study including ankle-brachial index (ABI) examinations and serum oxidative stress markers. The serum oxidative stress markers (high-sensitivity C-reactive protein (hs-CRP), matrix metalloproteinase-2 (MMP-2), MMP-9, homocysteine, asymmetrical dimethylarginine (ADMA), nitrate oxidase (NO), tumour necrosis factor-α (TNF-α), monocyte chemotactic protein 1 (MCP-1), interleukin-1β (IL-1β), and transforming growth factor-β (TGF-β)) were measured using immunosorbent assays in 159 HD patients (83 men and 76 women; mean age: 65 ± 12 years). The participants met the following criteria: (1) received regular HD treatment for at least 6 months, without clinical evidence of acute or chronic inflammation, recent myocardial infarction, unstable angina or circulatory congestion; and (2) received an arteriovenous fistula (AVF)/arteriovenous graft (AVG: polytetrafluoroethylene, PTFE) as the current VA for more than 6 months, without interventions within the last 6 months. All the participants were followed up clinically for up to 12 months to estimate the amount of primary thrombotic occlusion and VA dysfunction demanding PTA. During the 12-month observation, 24 patients (15.1%) had primary thrombotic occlusion of VAs. Another 24 patients (15.1%) required PTA because of clinical dysfunction of access. Additionally, during the follow-up period, restenosis occurred in 12 patients (50% of 24 patients). The access types of arteriovenous grafts (AVGs) and a diagnosis of peripheral arterial occlusive disease (PAOD) were two strong predictors for acute thrombotic events of VA (hazard ratio (HR): 16.93 vs. 2.35; *p* < 0.001 vs. 0.047). Comparing dysfunctional with non-dysfunctional VAs, up to 27.7% of patients with high levels of ADMA (>0.6207 μM, *N* = 65) received required PTA compared with 4.4% of those with low levels (≤0.6207 μM; *N* = 90; *p* < 0.001). In multivariate analysis, the plasma baseline levels of ADMA independently conferred nearly 4.55 times the risk of primary stenotic dysfunction of HD VA (HR: 4.55; 95% confidence interval: 1.20 to 17.26; *p* = 0.026). In conclusion, our findings suggest the role of ADMA in the development of symptomatic VA dysfunction. Additionally, PAOD severity can be used in clinical practice to predict whether acute thrombotic occlusion of VA will easily occur in HD patients.

## 1. Introduction

Durable and functional dialysis vascular access (VA) is important for every haemodialysis (HD) patient. VA dysfunction increases the morbidity and mortality in HD patients [1]. The most common cause of VA dysfunction is stenosis at the arteriovenous anastomosis due to progressive neointimal proliferation and extracellular matrix deposition. Intimal hyperplasia is produced by the proliferation of vascular smooth muscle cells (VSMCs), together with matrix deposition caused by the elaboration of many different growth factors and cytokines. These changes can also be observed in classic atheromas, indicating that atherosclerotic changes and venous stenosis in VA dysfunction may present comparable pathogenic mechanisms [2,3,4,5]. Currently, no well-established criteria or sensitive markers can predict acute thrombotic occlusion and progressive dysfunction of VA requiring percutaneous transluminal angioplasty (PTA) intervention in HD patients. If VA dysfunction cannot meet the requirements of each dialysis session, a high risk occurs for acute thrombotic occlusion. The patency rate of VA after PTA is attenuated by a high restenosis rate within 6 months [3,4,5,6]. Several previous experimental animal models have been used to elucidate the pathophysiology of primary to aggressive intimal hyperplasia [7,8,9].

Oxidative stress can be a significantly pathogenetic link to chronic kidney disease (CKD). Asymmetrical dimethylarginine (ADMA) is an endogenous inhibitor of nitric oxide (NO) synthase and a crucial contributor to endothelial dysfunction [10,11,12,13,14]. Matrix metalloproteinases (MMPs) are members of the zinc-dependent endopeptidase family produced by smooth muscle and endothelial cell fibroblasts and inflammatory cells [9]. Previous work in a porcine model of HD vascular failure showed increased MMP-2 expression with cellular migration from the adventitia and media, leading to venous stenosis formation [8]. MMP-2 and MMP-9 (also called gelatinase A and B, respectively), which are produced by both VSMCs and inflammatory cells, are the major MMPs associated with VSMC migration and neointimal formation after vascular injury [7]. High-sensitivity C-reactive protein (hs-CRP) and homocysteine are representative markers of oxidative stress from vascular injury and show a high correlation with coronary artery stenosis and restenosis after percutaneous coronary intervention [15]. Additionally, circulating levels of inflammatory cytokines, including tumour necrosis factor-α (TNF-α), monocyte chemotactic protein 1 (MCP-1), interleukin-1β (IL-1β), proliferative cytokines, and transforming growth factor-β (TGF-β), are markers of oxidative stress, which may be associated with VA dysfunction or VA failure [6,16,17,18,19,20].

To our best knowledge, it remains unclear whether these oxidative stress markers can be used as predictors for thrombotic occlusion of VA and progressive stenosis dysfunction demanding PTA. Therefore, in this pilot study we investigated the association of serum oxidative stress markers during vascular injury in HD patients. We prospectively compared whether a significant difference existed between VA with acute thrombosis and good function during the 12-month follow-up. Finally, we documented whether the risk of progressive dysfunction of VA demanding PTA could be evaluated using those markers.

## 2. Materials and Methods

### 2.1. Study Participants

One hundred sixty-five patients who were treated with HD were recruited to this study, and their data were collected at Armed Forces Taoyuan General Hospital, Taiwan, from February 2019 to January 2020 (Figure 1). The participants met the following criteria: (1) received regular dialysis treatment for at least 6 months, without clinical evidence of acute or chronic inflammation, recent myocardial infarction, unstable angina, or circulatory congestion; and (2) received an arteriovenous fistula (AVF)/arteriovenous graft (AVG: polytetrafluoroethylene, PTFE) as the current VA for more than 6 months, without interventions within the last 6 months. All the participants were followed up clinically for up to 12 months to estimate the amount of primary thrombotic occlusion and VA dysfunction demanding PTA.

The study was performed in accordance with the Helsinki Declaration (edition 6, revised in 2000). Informed consent was obtained from all the participants, and the study was approved by the Institutional Research Board of Tri-Service General Hospital, National Defense Medical Center, Taiwan. The approval number was TSGHIRB-09801 and TSGHIRB-C202005187.

### 2.2. Study Protocol

We recorded the following characteristics of the participants: age, sex, HD duration, underlying cause of uraemia (presence or absence of diabetic nephropathy), and time of VA creation. Baseline blood samples were collected after a 12-h overnight fast and withdrawal of medications. Cigarette smoking and the consumption of beverages containing alcohol or caffeine were also avoided for at least 12 h.

Primary end point (E1): The primary end point was the occurrence of acute thrombosis of VA. The survival time of the VA with unassisted patency was defined as the time from commencement of the study to an episode of thrombotic occlusion, which was proven by surgical thrombectomy [16].

Secondary end point (E2): Patients who experienced VA dysfunction requiring PTA to anatomically correct the abnormality.

The definition of dysfunction of VA is based on the rationale of the 2006 National Kidney Foundation, which issued the Kidney Disease Outcomes Quality Initiative (KDOQI) Clinical Practice Guidelines (CPGs) for Vascular Access: Guideline 4 (CPG 4): DETECTION OF ACCESS DYSFUNCTION assessed by clinical or physical abnormality (Monitoring, Table-17) [21]: decreased thrill, increased pulsatility, development of collateral veins, limb swelling, and prolonged bleeding from puncture sites with one or more of the following criteria (Surveillance): reduction in the flow rate of >25% from the baseline access flow, total access blood flow rate of less than 600 mL/min in the graft and less than 400–500 mL/min in the fistula by the ultrasound dilution method (Transonic Flow-QC; Transonic Systems, Ithaca, NY, USA), and increased venous pressure during dialysis (dynamic venous pressure exceeding the threshold levels three consecutive times) [21].

During the one-year follow-up, patients were withdrawn from the study with the following censoring criteria: death with functional access, shifting modality of renal replacement therapy (either peritoneal dialysis or renal transplantation), or loss to follow-up.

### 2.3. Laboratory Methods

To avoid any confounded effect from HD, a 20-mL blood sample was drawn from each patient before haemodialysis in the morning. Blood sampling was performed after 30 min of quiet rest in a semirecumbent position. The blood samples were centrifuged at 3000 rpm for 10 min at 4 °C immediately after collection. The plasma samples were then stored at −80 °C until use.

Plasma biochemical parameters, including low-density lipoprotein cholesterol (LDL-C), high-density lipoprotein cholesterol (HDL-C), triglycerides, calcium, phosphate, and albumin were analysed. NO was assayed in the blood sample using a sensitive and specific chemiluminescence detection method. After a series of steps to collect the total amount of plasma NO, blood samples were drawn into a Sievers Nitric Oxide Analyser (Sievers NOA 280i; Boulder, CO, USA). Blood samples were collected in EDTA-containing tubes and then centrifuged at 2000× *g* for 15 min at 4 °C. The plasma was frozen and stored at −80 °C until analysis. Plasma TNF-α, MCP-1, IL-1β, and TGF-β levels were measured using the enzyme-linked immunosorbent assay (ELISA) with commercially available kits (human tumour necrosis factor-alpha (TNF-α) ELISA Kit, cat. no. Ab181421, Abcam; human monocyte MCP-1 ELISA Kit, cat. no. Ab179886, Abcam; human IL-1β ELISA Kit, cat. no. Ab46052 Abcam; human TGF-β ELISA Kit, cat. no. Ab100647 Abcam) according to the manufacturer’s instructions. The plasma hs-CRP assay was based on the latex agglutination immunoassay method. When an antigen-antibody reaction occurs between CRP in a sample and anti-CRP antibody, which has been sensitized to latex particles, agglutination results. This agglutination is detected as an absorbance change (572 nm), with the magnitude of the change proportional to the quantity of CRP in the sample. The actual concentration is then determined by interpolation from a calibration curve prepared from calibrators of known concentration. The upper normal value of hs-CRP was 0.3 mg/dL in our laboratory. The plasma levels of ADMA were determined using commercially available enzyme-linked immunosorbent assay (ELISA) kits (DLD Diagnostika, Hamburg, Germany). The correlation coefficient between liquid chromatography-mass spectrometry ADMA and ELISA ADMA was 0.98. The recovery rate for ADMA was 95%, and the within-assay and between-assay variation coefficients were not more than 7% and 8%, respectively. The plasma levels of homocysteine were measured using the fluorescence polarization immunoassay based on the highly selective enzymatic conversion of homocysteine to *S*-adenosyl-l-homocysteine, which was then recognized by a monoclonal antibody (Abbott AxSYM Homocysteine; 5F51-20, Abbott Laboratory Abbott Park, Chicago, IL, USA). The default result unit for AxSYM homocysteine is μmol/L. The plasma levels of MMP-2 and MMP-9 were determined using commercially available ELISA kits according to the manufacturer’s instructions (Amersham Bioscience, Uppsala, Sweden). All the procedures were performed according to the manufacturer’s instructions. Each standard and each plasma sample were analysed twice, and the mean values were used for all subsequent data analyses.

Peripheral arterial occlusive disease (PAOD) testing was performed using a Nicolet VasoGuard physiological testing system (VasoGuard, P84/1249, UK) and the four-cuff method for segmental pressure and pulse volume recordings. Proximal vessels were selected using a 4 MHz transducer. Distal vessels were selected using an 8-MHz transducer for continuous wave Doppler. The ankle–brachial index (ABI) is a reliable marker for PAOD [22]. The ABI and toe-brachial index (TBI) were recorded in this study. The patients’ lower limbs were divided into two groups: those with an ABI greater than 0.9 and 1.3 or lower and those with an ABI of 0.90 or lower. We censored the abnormal remainder because the PAOD group had at least unilateral or bilateral leg ABI values of 0.9 or low. The prevalence of PAOD in the study population was 19%.

### 2.4. Follow-Up and Definitions

After the baseline blood collection in this study, all the participants were prospectively followed up for one year under the same protocol at their HD centre. Follow-up monitoring and surveillance included physical examination and dynamic venous pressure monitoring at each HD session. According to the KDOQI Work Group recommendation on Guideline Statements 15.1 [23], local resources and the severity of findings on clinical monitoring should determine the timeframe, choice, and extent of imaging studies for further evaluation of which timeframe of less than 2 weeks was deemed reasonable periodicity. The sudden onset of acute thrombotic occlusion of the VA was identified before each dialysis session and was confirmed by surgical thrombectomy. When abnormal clinical or haemodynamic parameters fulfilled the KDOQI Clinical Practice Guideline for VA: 2019 Update [23]—an indication for preemptive PTA—these patients were referred for fistulography and PTA as appropriate. Diagnostic fistulography and PTA were performed using standard procedures. After diagnostic fistulography or PTA, the patients with insignificant stenosis (<50% diameter stenosis), arterial side stenosis, or central vein lesions were excluded. The terms, definitions, and categories of features were in accordance with the Committee on Reporting Standards of the Society for Vascular Surgery and American Association for Vascular Surgery.

### 2.5. Statistical Analysis

Data management and statistical analysis were performed using SPSS statistical software (version 22; Chicago, IL, USA). The distributions of continuous variables in groups were expressed as means ± SD and were compared by the unpaired *t* test and Mann–Whitney U test for non-normally distributed data. Categorical variables were analysed by the χ2 test with Yates correction and Fisher’s exact test as appropriate. To establish a cut-off point between low and high levels, we related the percentiles of ADMA values and rate of dysfunction-free survival using a receiver operating characteristic (ROC) curve. Cumulative survival unassisted patency of HD VA (including fistula and graft) was estimated using the Kaplan–Meier method and was compared by the log-rank test. We performed univariate and multivariate Cox proportional hazards regression analyses to determine independent predictors for acute thrombotic occlusion and progressive dysfunction of VA. Adjusted variables with *p* < 0.05 by univariate analysis were included in the multivariate model. Hazard ratios (HRs) and 95% confidence intervals were calculated. A *p*-value <0.05 was considered statistically significant.

## 3. Results

### 3.1. Patient Characteristics

One hundred sixty-five patients with functional VA were enrolled in the study (Figure 1). Among them, six patients were excluded: two because of the use of a Hickman catheter with immature VA and four because of receiving the VA intervention procedure within the last 6 months of the start of this project. Therefore, the study group comprised 159 patients (83 men and 76 women; mean age: 65 ± 12 years). Twenty-four patients had acute venous thrombosis. The cumulative incidence of acute venous thrombosis in the study was 24 (24/159 = 15.09%).

### 3.2. Clinical Follow-Up

Blood samples were taken from each patient for baseline parameters. The time interval between cessation of the last dialysis and baseline blood sampling usually varied from 18 to 24 h. All the patients received one year of clinical follow-up. Twenty-four patients had sudden-onset thrombotic occlusion (15%, end-point 1, E1), and 24 had clinical findings indicating dysfunctional VA that met the criteria of the KDOQI clinical practice guidelines for VA requiring intervention with a PTA procedure (15%, end-point 2, E2). PTA was performed successfully on all the patients without major complications. During the follow-up of the patients with E2, 12 patients had recurrence (50%) meeting dysfunctional criteria that required repeat PTA (*n* = 7) or thrombotic failure (*n* = 5). Ultimately, no patient was lost to follow-up apart from 17 patients who die with functional accesses, 5 patients who died with dysfunctional accesses, and 6 patients who died after the diagnosis of thrombotic occlusion. Thus, 131 patients completed the one-year study. We calculated the following two-group comparisons: acute thrombotic VA occlusion patients versus well-functioning VA patients and clinically dysfunctional VA patients versus non-dysfunctional VA patients. The comparison of the study participants revealed no differences in the access blood flow rate in each group.

### 3.3. Baseline Parameters and Acute Thrombotic Occlusion

The characteristics of the 135 non-thrombotic VA subjects and 24 subjects with sudden-onset thrombotic occlusion of VA are listed in Table 1. The mean period of follow-up for the cumulative delivery of acute thrombotic occlusion was 3.67 ± 2.66 months. Statistical analysis revealed a significant difference in the access types with AVG (thrombosis (−) versus thrombosis (+), 23.7% versus 87.5%, *p* < 0.001); history of PAOD (thrombosis (−) versus thrombosis (+), 14.8% versus 41.7%, *p* = 0.005); serum level of phosphate (thrombosis (−) versus thrombosis (+), 4.69 ± 1.52 versus 5.38 ± 1.67 mg/dl, *p* = 0.043) and medication with aspirin (thrombosis (−) versus thrombosis (+), 40.7% versus 75%, *p* = 0.004). No differences were found in the plasma biomarkers: hs-CRP, homocysteine, ADMA, MMP-2, MMP-9, NO, MCP-1, TGF-β, TNF-α, and IL-1β. Additionally, no difference was found in traditional risk factors: age, history of hypertension, and type 2 diabetes mellitus between these groups.

In the univariate Cox regression analysis, access type, history of PAOD, and medication with aspirin were associated with an increased risk for symptomatic acute thrombotic occlusion of VA. Next, we performed multivariate Cox regression analysis to identify the independent predictors. Only access type of AVG (HR: 16.93, *p* < 0.001) and history of PAOD (HR: 2.35, *p* = 0.047) were independent predictors of acute thrombosis risk in our HD patients (Table 2). Further analysis after combining these two factors—namely, VA graft type and PAOD—revealed an acute thrombotic event rate of up to 50%-60% in our patients during the one-year follow-up (Figure 2).

### 3.4. Baseline Parameters and Progressive Dysfunction of Vascular Access

Twenty-four patients had dysfunction of dialysis VA (which fulfilled the 2006 DOQI clinical practice guidelines for VA) and required referral for evaluation and treatment (PTA) [21]. During this one-year prospective follow-up period, the cumulative incidence of the dysfunction was 15.09% (24/159). Of the patients 50% (12/24) experienced restenosis after the PTA procedure at the same location. The other 12 patients with the subsequent well function of VA did not receive PTA again. The mean period of follow-up for the cumulative delivery of progressive dysfunction was 4.18 ± 3.08 months. However, four non-restenosis patients were lost to follow-up owing to death from infectious or cardiac diseases (Table 3).

Compared with the 135 prospective follow-up subjects with clinical non-dysfunctional VA (dysfunction (−) 0.5642 ± 0.2426 μM), the 24 patients with progressive dysfunction had a significantly higher baseline plasma ADMA level (dysfunction (+) 0.774 ± 0.2735 μM, *p* = 0.001) and higher baseline MMP-9 level (dysfunction (+) 81.87 ± 47.16 ng/mL, *p* = 0.021). The TNF-α level in dysfunction subjects (dysfunction (+) 14.09 ± 37.00 pg/mL) had a significantly lower baseline plasma value than the patients with non-dysfunction (dysfunction (−) 40.88 ± 85.86 ng/mL, *p* = 0.016). There was no significant difference in age, history of hypertension and type 2 diabetes mellitus between these groups.

In univariate Cox regression analysis, access type, high serum ADMA levels, high MMP-9 levels, medication with diuretics, and lipid-lowering agents with statins were associated with an increased risk for symptomatic dysfunction of VA. Next, we performed multivariate Cox regression analysis to identify the independent predictors. Only plasma ADMA independently predicted whether progressive dysfunction of access occurred easily in our HD patients (Table 4; HR: 4.36; *p* = 0.031).

Based on the receiver operating characteristic analysis, a baseline plasma ADMA level of 0.6207 μmol/mL was identified as the best cut-off value for predicting symptomatic dysfunction of access (Figure 3). The patients were divided into high (>0.6207 μM) versus low (≤0.6207 μM) ADMA groups (shown in Table 5). In the high ADMA group, 27.7% of patients had VA dysfunction requiring PTA during the follow-up period compared with 4.4% in the low ADMA group (*p* < 0.001). Kaplan–Meier analysis showed that high (>0.6207 μM) ADMA was associated with significantly lower sustained access function (*p* < 0.001), i.e., dysfunction with higher ADMA (Figure 4).

Since access type had a significant impact on the VA outcome, subgroup analysis was performed as shown in Appendix A (acute thrombosis) and Appendix A (dysfunction). After multivariate analysis, no significance was found between plasma biomarkers and acute thrombosis or dysfunction of VA among the AVG groups. Among the AVF groups, homocysteine showed significance with acute thrombosis (HR: 1.08; *p* = 0.004); however, ADMA played an important role in predicting progressive stenotic dysfunction (HR: 28.93; *p* = 0.001). Furthermore, the correlation matrix between clinical factors and oxidative stress markers is shown in Appendix A, which also revealed the high correlation between ADMA and the dysfunction of AVF.

## 4. Discussion

Complications associated with established VA sites are important causes of morbidity and mortality in patients receiving regular HD. The most frequent complication leading to failure of the VA site is thrombosis [1]. The likelihood of thrombosis depends on multiple factors, including the development of myointimal hyperplasia, anatomic configuration of the fistula or graft constructed, site of arteriovenous anastomosis, selection of prosthetic material, and intrinsic clotting ability; however, the adequacy of the patient’s veins and arteries is probably most important. Thrombosis of shunts may occur soon after surgery or in the follow-up period. Early thrombosis, defined as occurring within the first month after placement, is often caused by technical factors, whereas later thrombosis, which occurs from 1 month on, is generally caused by venous run-off stenosis (neointimal hyperplasia), continued trauma to the VA site by needle puncture for HD, external pressure on the graft, hypotension, or central venous thrombosis. This article is the first to prospectively speculate on the correlation between endothelial injury markers and acute VA thrombotic events.

Prospective monitoring and surveillance of fistulae and grafts for haemodynamically significant stenosis, when combined with correction of anatomic stenosis, may improve patency rates and decrease the incidence of thrombosis [21,23]. Our study findings showed that, in plasma, ADMA is an important independent predictor of symptomatic dysfunction of VA that demands PTA intervention. In particular, patients with plasma ADMA levels >0.6207 μM were identified as having a higher VA dysfunction rate. Our results indicate the connection between baseline plasma ADMA and the consequent development of symptomatic access dysfunction and suggest the potential pathogenic role of these markers and endothelial dysfunction in predicting the progressive dysfunction of VA in our clinical HD patients. Failure to detect access dysfunction has consequences on morbidity and mortality. A recent meta-analysis reported that far-infrared therapy improved the endothelial function and access flow of newly created fistulas in HD patients [24]. Wu et al. [11] recommended the role of plasma ADMA in the restenosis process after PTA of AVF. Previous studies have declared the role of ADMA in interfering with the production of NO, which may be pivotal for venous dilatation in our HD patients [10,11,12]. Likewise, we can infer from our results that ADMA is important for VA patency. The results showed that the plasma level of ADMA might serve not only as a predictor of AVF maturation and restenosis after PTA but also as a promising predictor of VA dysfunction. ADMA is likely involved in endothelial dysfunction through competition with L-arginine as the substrate for NO synthase, resulting in decreased endothelium-dependent NO production. The endothelium plays a crucial role in maintaining vascular homeostasis, and NO is the most important mediator of this process [11]. The deprivation of NO production may contribute to leukocyte adhesion, smooth muscle proliferation, and extracellular matrix formation and accumulation because of the impaired renal clearance of ADMA in patients with uraemia [12]. This finding might explain why ADMA is a more significant determinant factor in dysfunction resulting from stenosis of HD accesses.

IL-1β, another major factor specific to AVF, can also induce inflammation not only because of AVF creation but also because of repeat needle stick injury. Platelets can adhere to the already inflamed endothelial tissue and potentiate the process by releasing IL-1β, MCP-1, TNF-α, and other inflammatory cytokines. These cytokines can further cause the activation cascade leading to increased inflammation, adhesion, and eventually plaque or thrombus formation [25,26,27]. Our study showed that a higher level of ADMA is associated with a higher level of MMP-9 and IL-1β, indicating inflammation and endothelial dysfunction can promote stenosis dysfunction of VA and lead to shorter duration of VA use. The ADMA level is significantly related to higher levels of LDL and homocysteine, suggesting the clear relationship to atherosclerosis and cardiovascular events, which could drive further progression of CKD and consequently more diuretics used. ADMA can regulate endothelial NO production by modulating the calcium-sensing receptor, thus increasing intracellular calcium release. This finding may also explain the higher level of serum calcium among patients with a higher ADMA level in this study [28]. TNF-α, a 17-kD protein, is a prominent inflammatory cytokine of interest in intimal hyperplasia. Unexpectedly, our study showed a lower level of TNF-α in patients with a higher level of ADMA. TNF-α has apoptotic properties, which can be masked by the antiapoptotic NF-kB pathway. Consequently, only when the NF-kB pathway and/or protein synthesis are non-functional does TNF-α become apoptotic. This finding may also explain why TNF-α usually appears to be proliferative with intimal hyperplasia [16]. Although many cells may show cellular injury at the time of AVF placement, it is insufficient to compromise the antiapoptotic effects. Thus, TNF-α in the setting of intimal hyperplasia can contribute synergistically with the proliferative effects of other cytokines released during AVF placement [25,26,27]. In our study, the sample size was small and the finding might require further delineation.

The present investigation was a pilot study with a small sample size in addition to the VA type used, and PAOD comorbidity is another strong predictor in acute thrombotic VA. PAOD is prevalent in patients with an advanced stage of CKD, but many factors related to a low or high pathological ABI differ, revealing different pathogenic pathways. Among end-stage renal disease (ESRD) patients, diabetes mellitus, chronic inflammation, mineral and bone disorders, and dyslipidaemia may also play a role in the appearance of PAOD [29,30,31]. PAOD is mainly caused by atherosclerosis of the vessel wall, which may be associated with VA dysfunction in HD patients [5,18,19,20]. In our study, the risk of acute thrombosis occlusion in AVG with PAOD was higher, significantly approaching 60%. Nevertheless, most of oxidative stress markers did not influence the incidence of acute-onset thrombosis in our study. In contrast to plaque rupture resulting in the development of thrombosis in arterial systems, no correlation was found between VA thrombosis among HD patients and traditional cardiovascular risk factors or anatomic factors, except for the observed poor patency. However, in many of these patients, the causes of rapid progression and individual variation in the development of acute thrombosis remain unknown. In practice, nephrologists may overlook the presentation of PAOD and consider it a problem for cardiovascular surgery. If we are aware of its presentation, particularly in the VA type, which occurs in grafts, the possibility of surgical thrombectomy can be attenuated. Our prospective study showed no significance between plasma biomarkers and acute thrombosis or dysfunction among the AVG groups. Arterial vascular disease is predominantly characterized by atherosclerosis with various inflammatory components [12]. However, the venous segment of VA is characterized by neointimal hyperplasia, which differs from characteristic atherosclerotic lesions. Among the AVF groups, ADMA seemed to affect the prediction of progressive stenotic dysfunction (Appendix A) in our study. Homocysteine also showed significance with acute thrombosis (Appendix A) among the AVF groups. Confirmation that ADMA is a powerful predictor of the rate of progressive dysfunction of AVF would raise the possibility of a novel means of risk stratification and new therapeutic approaches.

These findings should be evaluated considering the following limitations in our study. First, our patients were followed up mainly for clinical symptoms and signs suggesting VA dysfunction; therefore, we cannot exclude the possible presence of silent VA problems that were not studied. Second, a substantial ethnic variation in the plasma ADMA levels has been reported, and this variation might hamper the application of our cut-off value to other populations. Third, our cohort was cross-sectional in design to estimate the primary assisted patency not from the initiation of each VA construction. There could be some intervention therapies before enrolment in our study. Fourth, we measured the levels of oxidative stresses at the beginning of this study, likely not necessarily representing the levels during the whole follow-up period. It remains unknown whether those marker levels and other potential indicators may change with time, and whether such changes may contribute to the development of VA dysfunction [25,26,27]. Fifth, for standardization of the analysis, ELISA was used in our study for ADMA determination, instead of the more precise mass spectrometric method. We cannot completely exclude that ELISA may detect other factors that are responsible for the observed difference. Regarding the ADMA biomarkers, the comparison between the UPLC-MS/MS method and ELISA showed only a moderate correlation. Therefore, this finding must be considered with absolute concentrations [32]. Finally, no evidence is available that modifies the PAOD severity or ADMA levels will reduce the rate of acute thrombosis or dysfunction of VA.

## 5. Conclusions

Our pilot study presented encouraging results but had a small sample size and this cohort was cross-sectional in design, estimating the primary assisted patency not from the initiation of each VA construction of HD patients to understand which factors was statistical significance in HD VA. We found that clinically manifested PAOD, in addition to AVG in ESRD patients, is an independent determinant of acute thrombotic occlusion. The plasma marker ADMA is a predictor of VA dysfunction in HD patients and a precursor of acute thrombosis. Additional research to define PAOD in relation to thrombotic occlusion and the role of ADMA in the pathogenetic mechanisms of VA are warranted.

## Figures and Tables

**Figure 1 antioxidants-10-00569-f001:**
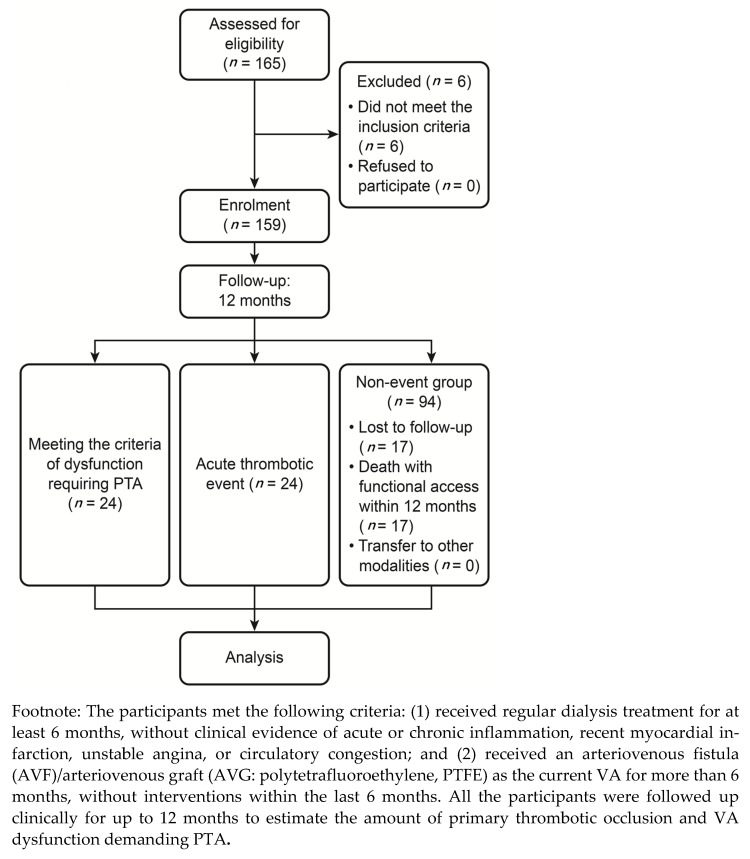
Flow chart of the study participants in prospective cohort trials to evaluate biomarkers and survival either between thrombotic and non-thrombotic haemodialysis (HD) patients or between progressive dysfunction demanding percutaneous transluminal angioplasty (PTA) and intact maintenance function of vascular access (VA).

**Figure 2 antioxidants-10-00569-f002:**
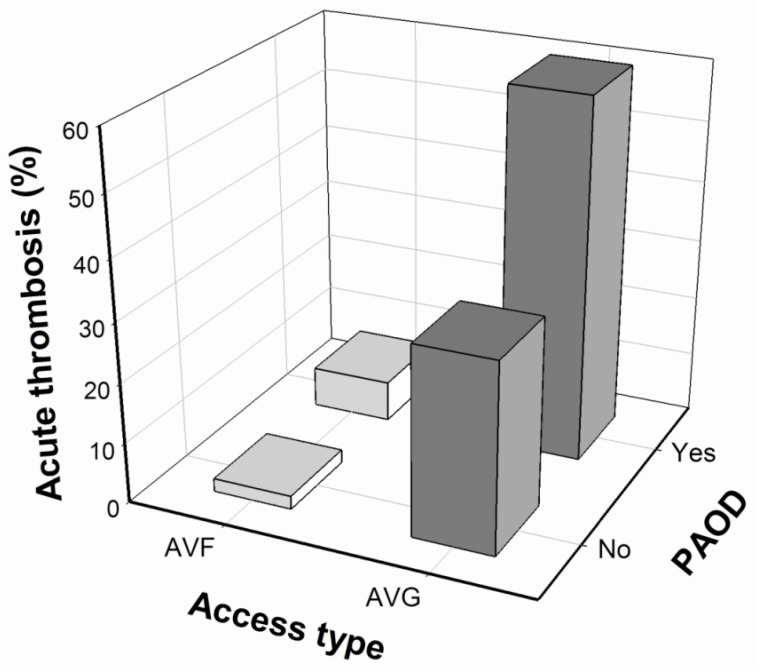
Relationship among the risk of acute thrombosis occlusion in VA, access type and PAOD in patient maintenance haemodialysis. The variables AVF and AVG represent the arteriovenous fistula and arteriovenous graft, respectively (PAOD: peripheral artery occlusive disease).

**Figure 3 antioxidants-10-00569-f003:**
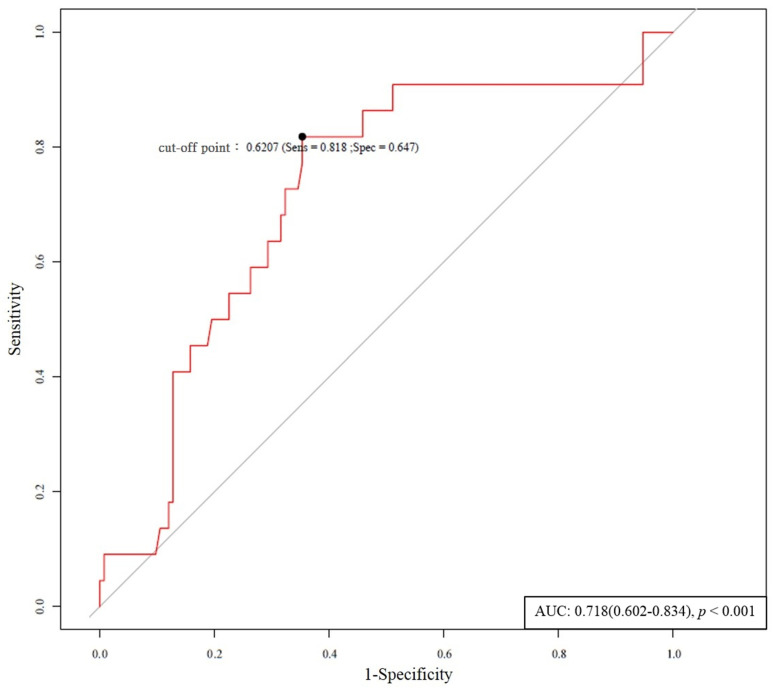
Receiver operating characteristic (ROC) curve of ADMA values for the rate of dysfunction-free survival and provide the best cut-off point.

**Figure 4 antioxidants-10-00569-f004:**
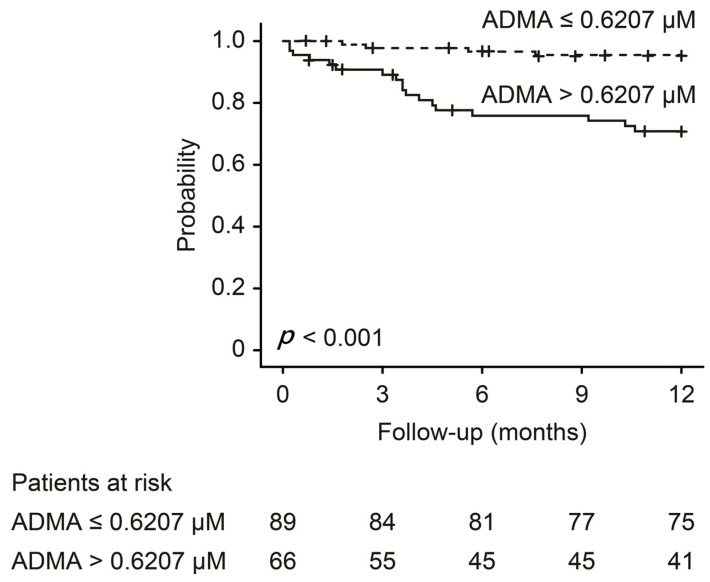
Kaplan–Meier analyses of the proportion of patients without dysfunctional vascular access. The patients were divided according to a cut-off baseline ADMA value of 0.6207 μM.

**Table 1 antioxidants-10-00569-t001:** Baseline characteristics of the study patients (total *N*: 159) according to acute thrombosis of access vs. patency of access throughout the 12-month prospective follow-up.

Characteristics	Acute Thrombosis (*n* = 24)	Non-Thrombosis (*n* = 135)	*p*
Age (years), mean ± SD	68.38 ± 12.08	64.33 ± 12.77	0.145
Sex (%)			
Female	13 (54.2%)	63 (46.7%)	0.648
Male	11 (45.8%)	72 (53.3%)	
Access type (%)			
AVF	3 (12.5%)	103 (76.3%)	<0.001 *
AVG	21 (87.5%)	32 (23.7%)	
Duration of access use (months), mean ± SD	3.67 ± 2.68	10.91 ± 2.83	<0.001 *
Risk factors (%)			
Hypertension	20 (83.3%)	109 (80.7%)	0.987
Diabetes mellitus	13 (54.2%)	58 (43.0%)	0.427
Current smoking	1 (4.2%)	12 (8.9%)	0.693
CAD	13 (54.2%)	46 (34.1%)	0.099
PAOD	10 (41.7%)	20 (14.8%)	0.005 *
Plasma biochemical data, mean ± SD			
LDL-C (mg/dL)	96.67 ± 35.30	90.75 ± 34.23	0.439
HDL-C (mg/dL)	44.38 ± 14.03	42.49 ± 15.92	0.406
TG (mg/dL)	189.92 ± 150.24	152.36 ± 89.78	0.437
Calcium (mg/dL)	9.15 ± 0.99	9.27 ± 1.02	0.597
Phosphate (mg/dL)	5.38 ± 1.67	4.69 ± 1.52	0.043 *
Albumin (mg/dL)	3.80 ± 0.42	3.85 ± 0.33	0.530
Creatinine (mg/dL)	10.06 ± 3.22	10.44 ± 2.57	0.519
Kt/V	1.35 ± 0.22	1.41 ± 0.29	0.359
Medications (%)			
Anti-platelet agents	19 (79.2%)	69 (51.1%)	0.020 *
Aspirin	18 (75.0%)	55 (40.7%)	0.004 *
Dipyridamole	3 (12.5%)	6 (4.4%)	0.321
Cilostazol	5 (20.8%)	16 (11.9%)	0.137
Coumadin	1 (4.2%)	4 (3.0%)	0.564
Nitrates	10 (41.7%)	54 (40.0%)	1.000
β-blockers	8 (33.3%)	48 (35.6%)	1.000
Calcium antagonists	16 (66.7%)	74 (54.8%)	0.392
ACEI/ARB	9 (37.5%)	37 (27.4%)	0.447
Diuretics	9 (37.5%)	39 (28.9%)	0.545
Lipid-lowering agents			
Statin	3 (12.5%)	12 (8.9%)	0.703
Fibrate	1 (4.2%)	3 (2.2%)	0.484
Plasma biomarkers, mean ± SD			
Hs-CRP (mg/dL)	0.94 ± 1.15	1.15 ± 2.12	0.255
Homocysteine (μmol/L)	29.984 ± 15.00	26.92 ± 10.44	0.319
ADMA (μmol/mL)	0.67 ± 0.28	0.58 ± 0.25	0.102
MMP-2 (ng/mL)	760.28 ± 180.73	855.10 ± 208.99	0.053
MMP-9 (ng/mL)	62.12 ± 41.57	62.53 ± 41.01	0.886
NO (μM)	219.39 ± 196.25	257.23 ± 299.59	0.436
MCP-1 (pg/mL)	319.91 ± 118.32	338.64 ± 177.82	0.635
TGF-β (pg/mL)	3.23 ± 4.79	6.09 ± 8.25	0.117
TNF-α (pg/mL)	20.09 ± 66.09	39.64 ± 82.99	0.296
IL-1β (pg/mL)	66.91 ± 43.14	57.47 ± 44.02	0.352

*: *p*-value < 0.05; abbreviations: ACEI, angiotensin-converting enzyme inhibitor; ADMA, asymmetrical dimethylarginine; ARB, angiotensin receptor blocker; AVF, arteriovenous fistula; AVG, arteriovenous graft; CAD, coronary artery disease; HDL-C, high-density lipoprotein cholesterol; Hs-CRP, high-sensitivity C-reactive protein; LDL-C, low-density lipoprotein cholesterol; MMP-2, matrix metalloproteinase-2; MMP-9, matrix metalloproteinase-9; NO, nitrate oxidase; PAOD, peripheral arterial occlusive disease; TG, triglyceride; IL-1β: interleukin-1β; MCP-1: monocyte chemotactic protein 1; TNF-α: tumour necrosis factor-α; TGF-β: transforming growth factor-β.

**Table 2 antioxidants-10-00569-t002:** Univariate and multivariate Cox regression analyses for the predictors of acute thrombosis of arteriovenous fistulas/grafts within 12 months of follow-up.

Parameter	Univariate Analysis		Multivariate Analysis	
	HR (95% Cl)	*p*	HR (95% Cl)	*p*
Age (years)	1.03 (0.996–1.07)	0.084		
Sex				
Female	1.00			
Male	0.75 (0.33–1.66)	0.472		
Access type (AVF/AVG)				
AVF	1.00			
AVG	19.05 (5.67–63.998)	<0.001 *	16.93 (3.828–50.935)	<0.001 *
Risk factors (%)				
Hypertension	1.17 (0.40–3.43)	0.773		
Diabetes mellitus	1.56 (0.697–3.47)	0.281		
Current smoking	0.47 (0.6–3.49)	0.461		
CAD	2.16 (0.97–4.83)	0.060		
PAOD	3.63 (1.61–8.18)	0.002 *	2.35 (0.901–6.476)	0.047
Plasma biochemical data				
LDL-C > 130 mg/dL	0.87 (0.26–2.93)	0.826		
HDL-C < 50 mg/dL	1.00 (0.398–2.52)	0.997		
TG > 200 mg/dL	1.15 (0.46–2.89)	0.769		
Calcium (mg/dL)	0.92 (0.61–1.38)	0.687		
Phosphate (mg/dL)	1.32 (1.01–1.71)	0.043 *	1.266 (0.945–1.696)	0.114
Albumin (mg/dL)	0.56 (0.17–1.83)	0.334		
Creatinine (mg/dL)	0.92 (0.78–1.08)	0.290		
Kt/V	0.39 (0.08–1.92)	0.248		
Medications				
Anti-platelet agents	3.26 (1.22–8.74)	0.019 *	0.741 (0.078–6.994)	0.793
Aspirin	3.78 (1.50–9.52)	0.005 *	2.172 (0.274–17.202)	0.462
Dipyridamole	1.88 (0.70–5.04)	0.209		
Cilostazol	3.19 (0.95–10.69)	0.061		
Coumadin	1.53 (0.21–11.32)	0.678		
Nitrates	1.15 (0.51–2.60)	0.732		
β-blockers	0.89 (0.38–2.08)	0.790		
Calcium antagonists	1.54 (0.66–3.59)	0.321		
ACEI/ARB	1.61 (0.70–3.67)	0.261		
Diuretics	1.37 (0.6–3.14)	0.453		
Lipid-lowering agents				
Statin	1.30 (0.39–4.34)	0.675		
Fibrate	1.88 (0.25–13.95)	0.537		
Plasma biomarkers				
Hs-CRP > 0.5 mg/dL	1.61 (0.72–3.58)	0.248		
Homocysteine (μmol/L)	1.02 (0.99–1.05)	0.189		
ADMA (μmol/mL)	3.45 (0.78–15.14)	0.101		
MMP-2 (ng/mL)	0.998 (0.006–1.00)	0.045 *	0.998 (0.996–1.001)	0.142
MMP-9 (ng/mL)	1.00 (0.99–1.01)	0.932		
NO (μM)	0.999 (0.997–1.00)	0.493		
MCP-1 (pg/mL)	0.999 (0.996–1.002)	0.594		
TGF-β (pg/mL)	0.936 (0.858–1.021)	0.138		
TNF-α (pg/mL)	0.996 (0.988–1.004)	0.310		
IL-1β (pg/mL)	1.004 (0.995–1.013)	0.390		

*: *p*-value < 0.05; multivariate analysis: adjusted variables with significance (*p* < 0.05) in univariate analysis. HR was presented with 95% CI (range). Abbreviations: ACEI, angiotensin-converting enzyme inhibitor; ADMA, asymmetrical dimethylarginine; ARB, angiotensin receptor blocker; AVF, arteriovenous fistula; AVG, arteriovenous graft; CAD, coronary artery disease; HDL-C, high-density lipoprotein cholesterol; Hs-CRP, high-sensitivity C-reactive protein; LDL-C, low-density lipoprotein cholesterol; MMP-2, matrix metalloproteinase-2; MMP-9, matrix metalloproteinase-9; NO, nitrate oxidase; PAOD, peripheral arterial occlusive disease; TG, triglyceride; IL-1β: interleukin-1β; MCP-1: monocyte chemotactic protein 1; TNF-α: tumour necrosis factor-α; TGF-β: transforming growth factor-β.

**Table 3 antioxidants-10-00569-t003:** Baseline characteristics of the study patients (total *N*: 159) according to dysfunction of access vs. patency of access throughout the 12-month prospective follow-up.

	Dysfunction (*n* = 24)	Non-Dysfunction (*n* = 135)	*p*
Age (years), mean ± SD	62.21 ± 10.48	65.42 ± 13.04	0.156
Sex (%)			0.648
Female	13 (54.2%)	63 (46.7%)	
Male	11 (45.8%)	72 (53.3%)	
Access type (%)			0.060
AVF	12 (50.0%)	94 (69.6%)	
AVG	12 (50.0%)	41 (30.4%)	
Duration of access use (months), mean ± SD	4.18 ± 3.08	11.1 ± 2.69	<0.001 *
Risk factors (%)			
Hypertension	21 (87.5%)	108 (80.0%)	0.572
Diabetes mellitus	12 (50.0%)	59 (43.7%)	0.727
Current smoking	2(8.3%)	11 (8.1%)	1.00
CAD	10 (41.7%)	49 (36.3%)	0.785
PAOD	3 (12.5%)	27 (20.0%)	0.572
Plasma biochemical data, mean ± SD			
LDL-C (mg/dL)	100.88 ± 24.75	89.98 ± 35.64	0.086
HDL-C (mg/dL)	43.96 ± 17.13	42.57 ± 15.39	0.902
TG (mg/dL)	152.29 ± 93.42	159.05 ± 103.17	0.916
Calcium (mg/dL)	9.57 ± 1.06	9.19 ± 1.00	0.058
Phosphate (mg/dL)	5.07 ± 1.58	4.74 ± 1.55	0.510
Albumin (mg/dL)	3.88 ± 0.34	3.83 ± 0.35	0.520
Creatinine (mg/dL)	11.07 ± 2.48	10.26 ± 2.69	0.186
Kt/V	1.38 ± 0.19	1.41 ± 0.30	0.778
Medications (%)			
Anti-platelet agents	15 (62.5%)	73 (54.1%)	0.444
Aspirin	11 (45.8%)	62 (45.9%)	0.993
Dipyridamole	1 (4.2%)	8 (5.9%)	1.00
Cilostazol	4 (16.7%)	17 (12.6%)	0.527
Coumadin	2 (8.3%)	3 (2.2%)	0.164
Nitrates	10 (41.7%)	54 (40.0%)	0.878
β-blockers	8 (33.3%)	48 (35.6%)	0.834
Calcium antagonists	13 (54.2%)	77 (57.0%)	0.794
ACEI/ARB	7 (29.2%)	39 (28.9%)	0.978
Diuretics	15 (62.5%)	33 (24.4%)	<0.001 *
Lipid-lowering agents			
Statin	5 (20.8%)	10 (7.4%)	0.054
Fibrate	0 (0.0%)	4 (3.0%)	1.00
Plasma biomarkers, mean ± SD			
Hs-CRP (mg/dL)	1.09 ± 1.76	1.13 ± 2.05	0.191
Homocysteine (μmol/L)	26.27 ± 6.53	27.58 ± 11.90	0.874
ADMA (μmol/mL)	0.7740 ± 0.2735	0.5642 ± 0.2426	0.001 *
MMP-2 (ng/mL)	793.63 ± 221.14	848.72 ± 204.57	0.336
MMP-9 (ng/mL)	81.87 ± 47.16	59.13 ± 39.04	0.021 *
NO (μM)	357.84 ± 516.03	232.62 ± 220.21	0.051
MCP-1 (pg/mL)	324.48 ± 106.73	337.96 ± 179.54	0.728
TGF-β (pg/mL)	5.83 ± 6.80	5.64 ± 8.10	0.918
TNF-α (pg/mL)	14.09 ± 37.08	40.88 ± 85.86	0.016 *
IL-1β (pg/mL)	73.17 ± 39.82	56.27 ± 44.22	0.089

*: *p*-value < 0.05; abbreviations: ACEI, angiotensin-converting enzyme inhibitor; ADMA, asymmetrical dimethylarginine; ARB, angiotensin receptor blocker; AVF, arteriovenous fistula; AVG, arteriovenous graft; CAD, coronary artery disease; HDL-C, high-density lipoprotein cholesterol; Hs-CRP, high-sensitivity C-reactive protein; LDL-C, low-density lipoprotein cholesterol; MMP-2, matrix metalloproteinase-2; MMP-9, matrix metalloproteinase-9; NO, nitrate oxidase; PAOD, peripheral arterial occlusive disease; TG, triglyceride; IL-1β: interleukin-1β; MCP-1: monocyte chemotactic protein 1; TNF-α: tumour necrosis factor-α; TGF-β: transforming growth factor-β.

**Table 4 antioxidants-10-00569-t004:** Univariate and multivariate Cox regression analysis for the predictors of dysfunction of arteriovenous fistulas/grafts within 12 months of follow-up.

Parameter	Univariate Analysis		Multivariate Analysis	
	HR (95% Cl)	*p*	HR (95% Cl)	*p*
Age (years)	0.98 (0.95–1.02)	0.312		
Sex		0.506		
Female	1.00			
Male	0.76 (0. 34–1.70)			
Access type (AVF/AVG)				
AVF	1.00		1.00	
AVG	2.28 (1.03–5.08)	0.043*	2.39 (0.91–6.28)	0.076
Risk factors				
Hypertension (%)	1.65 (0.49–5.53)	0.418		
Diabetes mellitus (%)	1.36 (0.61–3.04)	0.448		
Current smoking (%)	0.99 (0.23–4.20)	0.986		
CAD (%)	1.26 (0.56–2.83)	0.581		
PAOD (%)	0.57 (0.17–1.90)	0.357		
Plasma biochemical data				
LDL-C (mg/dL)	1.01 (0.99–1.02)	0.193		
HDL-C (mg/dL)	1.00 (0.98–1.03)	0.794		
TG (mg/dL)	1.00 (0.99–1.00)	0.735		
Calcium (mg/dL)	1.44 (1.00–2.10)	0.053		
Phosphate (mg/dL)	1.14 (0.87–1.48)	0.354		
Albumin (mg/dL)	1.17 (0.35–3.86)	0.799		
Creatinine (mg/dL)	1.09 (0.94–1.26)	0.258		
Kt/V	0.71 (0.16–3.13)	0.652		
Medications				
Anti-platelet agents	1.35 (0.59–3.07)	0.482		
Aspirin	0.96 (0.43–2.14)	0.918		
Dipyridamole	1.41 (0.48–4.12)	0.532		
Cilostazol	0.77 (0.10–5.67)	0.793		
Coumadin	3.06 (0.72–13.03)	0.13		
Nitrates	1.10 (0.49–2.47)	0.821		
β-blockers	0.89 (0.38–2.07)	0.777		
Calcium antagonists	0.81 (0.36–1.80)	0.603		
ACEI/ARB	0.99 (0.41–2.38)	0.973		
Diuretic	4.23 (1.85–9.66)	0.001 *	1.80 (0.63–5.16)	0.277
Lipid-lowering agents				
Statin	2.80 (1.05–7.50)	0.040 *	1.16 (0.35–3.83)	0.810
Fibrate	0.05 (<0.0001–3221.32)	0.592		
Plasma biomarkers				
Hs-CRP (mg/dL)	1.01 (0.83–1.42)	0.907		
Homocysteine (μmol/L)	0.99 (0.95–1.03)	0.634		
ADMA (μmol/mL)	13.90 (3.20 – 60.45)	<0.001 *	4.36 (1.14–16.65)	0.031 *^,a^
MMP-2 (ng/mL)	0.999 (0.997–1.001)	0.471		
MMP-9 (ng/mL)	1.01 (1.00–1.02)	0.011 *	1.00 (0.99–1.01)	0.602
NO (μM)	1.00 (1.000–1.002)	0.089		
MCP-1 (pg/mL)	1.00 (0.997–1.002)	0.711		
TGF-β (pg/mL)	1.01 (0.96–1.06)	0.860		
TNF-α (pg/mL)	0.99 (0.98–1.00)	0.204		
IL-1β (pg/mL)	1.01 (0.999–1.015)	0.086		

Footnote: ^a^: The baseline plasma ADMA level of 0.6207 μmol/mL was identified as the best cut-off value for predicting symptomatic dysfunction of access (Figure 3); therefore, ADMA > 0.6207 μM was used as an indicator of plasma biomarkers in Appendix A. In multivariate analysis, the plasma baseline levels of ADMA independently conferred nearly 4.55 times the risk of primary stenotic dysfunction of HD VA (HR: 4.55; 95% confidence interval: 1.20–17.26, *p* = 0.026) (shown in Appendix A). *: *p*-value < 0.05; multivariate analysis: adjusted variables with significance (*p* < 0.05) in univariate analysis. HR was presented with 95% CI (range). Abbreviations: ACEI, angiotensin-converting enzyme inhibitor; ADMA, asymmetrical dimethylarginine; ARB, angiotensin receptor blocker; AVF, arteriovenous fistula; AVG, arteriovenous graft; CAD, coronary artery disease; HDL-C, high-density lipoprotein cholesterol; Hs-CRP, high-sensitivity C-reactive protein; LDL-C, low-density lipoprotein cholesterol; MMP-2, matrix metalloproteinase-2; MMP-9, matrix metalloproteinase-9; NO, nitrate oxidase; PAOD, peripheral arterial occlusive disease; TG, triglyceride; IL-1β: interleukin-1β; MCP-1: monocyte chemotactic protein 1; TNF-α: tumour necrosis factor-α; TGF-β: transforming growth factor-β.

**Table 5 antioxidants-10-00569-t005:** Patients, characteristics, and events at follow-up according to the baseline plasma asymmetrical dimethylarginine (ADMA) level.

Characteristics	ADMA > 0.6207 μM (*n* = 65)	ADMA ≤ 0.6207 μM (*n* = 90)	*p*
Age (years), mean ± SD	64.29 ± 13.19	65.29 ± 12.61	0.758
Sex (%)			
Female	35 (53.8%)	38 (42.2%)	0.205
Male	30 (46.2%)	52 (57.8%)	
Access type (AVF/AVG)			
AVF	41 (63.1%)	63 (70.0%)	0.464
AVG	24 (36.9%)	27 (30.0%)	
Duration of access use (months)	9.02 ± 4.40	10.86 ± 2.90	0.001 *
Risk factors			
Hypertension (%)	57 (87.7%)	68 (75.6%)	0.093
Diabetes mellitus (%)	31 (47.7%)	39 (43.3%)	0.708
Current smoking (%)	6 (9.2%)	7 (7.8%)	0.977
CAD (%)	26 (40.0%)	32 (35.6%)	0.692
PAOD (%)	10 (15.4%)	19 (21.1%)	0.488
Plasma biochemical data			
LDL-C (mg/dL)	104.02 ± 35.37	80.93 ± 28.16	<0.001 *
HDL-C (mg/dL)	41.55 ± 15.23	43.76 ± 15.88	0.390
TG (mg/dL)	159.86 ± 116.27	156.99 ± 90.07	0.863
Calcium (mg/dL)	9.60 ± 0.88	8.98 ± 1.05	<0.001 *
Phosphate (mg/dL)	4.76 ± 1.64	4.83 ± 1.52	0.757
Albumin (mg/dL)	3.85 ± 0.39	3.82 ± 0.31	0.549
Creatinine (mg/dL)	10.58 ± 2.98	10.24 ± 2.48	0.549
Kt/V	1.44 ± 0.23	1.37 ± 0.32	0.170
Medications			
Anti-platelet agents	33 (50.8%)	52 (57.8%)	0.483
Aspirin	29 (44.6%)	41 (45.6%)	1.000
Dipyridamole	3 (4.6%)	5 (5.6%)	0.667
Cilostazol	7 (10.8%)	13 (14.4%)	1.000
Coumadin	4 (6.2%)	1 (1.1%)	0.196
Nitrates	28 (43.1%)	35 (38.9%)	0.720
β-blockers	27 (41.5%)	26 (28.9%)	0.142
Calcium antagonists	39 (60.0%)	47 (52.2%)	0.425
ACEI/ARB	26 (40.0%)	18 (20.0%)	0.011 *
Diuretics	37 (56.9%)	8 (8.9%)	<0.001 *
Lipid-lowering agents			
Statin	9 (13.8%)	4 (4.4%)	0.073
Fibrate	1 (1.5%)	3 (3.3%)	0.855
Plasma biomarkers			
Hs-CRP (mg/dL)	1.23 ± 2.29	1.06 ± 1.82	0.620
Homocysteine (μmol/L)	31.18 ± 13.37	24.45 ± 8.42	<0.001 *
MMP-2 (ng/mL)	823.46 ± 223.15	847.86 ± 188.02	0.462
MMP-9 (ng/mL)	87.69 ± 44.83	44.85 ± 27.08	<0.001 *
NO (μM)	280.37 ± 268.54	232.96 ± 302.88	0.315
MCP-1 (pg/mL)	336.32 ± 114.16	331.40 ± 200.91	0.865
TGF-β (pg/mL)	6.32 ± 9.74	5.38 ± 6.56	0.515
TNF-α (pg/mL)	12.46 ± 37.64	54.73 ± 97.68	<0.001 *
IL-1β (pg/mL)	90.48 ± 41.61	37.93 ± 32.18	<0.001 *
Events at follow-up (n %)	18 (27.7%)	4 (4.4%)	<0.001 *

*: *p*-value < 0.05; abbreviations: ACEI, angiotensin-converting enzyme inhibitor; ADMA, asymmetrical dimethylarginine; ARB, angiotensin receptor blocker; AVF, arteriovenous fistula; AVG, arteriovenous graft; CAD, coronary artery disease; HDL-C, high-density lipoprotein cholesterol; Hs-CRP, high-sensitivity C-reactive protein; LDL-C, low-density lipoprotein cholesterol; MMP-2, matrix metalloproteinase-2; MMP-9, matrix metalloproteinase-9; NO, nitrate oxidase; PAOD, peripheral arterial occlusive disease; TG, triglyceride; IL-1β: interleukin-1β; MCP-1: monocyte chemotactic protein 1; TNF-α: tumour necrosis factor-α; TGF-β: transforming growth factor-β.

## Data Availability

The data underlying this article will be shared upon reasonable request to the corresponding author.

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
