# Peer review of "The Role of Oxidative Stress Markers in Predicting Acute Thrombotic Occlusion of Haemodialysis Vascular Access and Progressive Stenotic Dysfunction Demanding Angioplasty"

_antioxidants, 2021, doi:10.3390/antiox10040569_

Round 1
Reviewer 1 Report
Chan et al. investigated the association between oxidative stress markers and acute thrombotic occlusion of vascular access or stenotic dysfunction in hemodialysis patients. The study is interesting, but several study concerns should be mentioned.
1. How and when to collect the blood samples from hemodialysis should be described in detail.
2. How to select the adjusted factors in a multivariable model? In Tables 2 and 4, only age and gender were adjusted in the multivariable analysis. In table 2, access type, PAOD, phosphate, anti-platelet agents, aspirin, and MMP-2 were significant in univariate analysis. In table 4, access type, diuretic, statin, ADMA, and MMP-9 were significant in univariate analysis. In general, we may control all other potential confounders in the multivariable model. Thus, why the author only select age and gender as confounders in this study?
3. In Table 4, the ADMA markers in the outcome of arteriovenous fistulas/grafts’ dysfunction were evaluated using dichotomous data but in the outcome of arteriovenous fistulas/grafts’ acute thrombosis using continuous data. Why the analysis process is different. Besides, other markers were all analyzed using continuous data. Thus, AMDA should be treated in the same way as other oxidative stress markers.
4. Please demonstrate the ROC curve of ADMA values for the rate of dysfunction-free survival and provide the best cut-off point in the figure.
5. Since access type had a significant impact on the outcome (acute thrombosis or AV dysfunction), a subgroup analysis to investigate the oxidative stress markers and outcome association stratifies the vascular access type.
6. As for the ADMA biomarkers, the comparison between the UPLC-MS/MS method and the ELISA showed only a moderate correlation. Therefore, this needs to be taken into account when considering absolute concentrations. A reference could be cited and discussed in the study limitation (Jente Boelaert et al. Toxins. 2016 May 13;8(5):149).
7. The author demonstrated table 5 to illustrate the patients' characteristics differences between high and low ADMA levels. However, a correlation matrix between clinical factors and oxidative stress markers would also be interesting to find the possible clinical effect on ADMA level in hemodialysis patients.
8. A higher level of ADMA were presented a higher VA dysfunction rate. Patients with a higher level of ADMA presented fewer duration of access use, more diuretics used, a higher level of LDL, calcium, homocysteine, MMP-9, and IL-beta, as well as a lower level of TNF-alpha. A brief discussion was recommended to explain these findings.
Author Response
Reviewer 1
Chan et al. investigated the association between oxidative stress markers and acute thrombotic occlusion of vascular access or stenotic dysfunction in hemodialysis patients. The study is interesting, but several study concerns should be mentioned.Author Reply: We sincerely appreciate your time and effort spent in reviewing this manuscript. We have revised the manuscript thoroughly according to your suggestions. The responses to your comments are found below. 1. How and when to collect the blood samples from hemodialysis should be described in detail.Author Reply: Thank you for the suggestion. We have amended the manuscript as follows: To avoid any confounded effect from haemodialysis, a 20-ml blood sample was drawn from each patient before haemodialysis (HD) in the morning. Blood sampling was performed after 30 min of quiet rest in a semirecumbent position. The blood samples were centrifuged at 3000 rpm for 10 min at 4°C immediately after collection. The plasma samples were then stored at -80℃ until use. Plasma biochemical parameters, including low-density lipoprotein cholesterol (LDL-C), high-density lipoprotein cholesterol (HDL-C), triglycerides, calcium, phosphate, and albumin, were analysed. NO was assayed in the blood sample using a sensitive and specific chemiluminescence detection method. After a series of steps to collect the total amount of plasma NO, blood samples were drawn into a Sievers Nitric Oxide Analyser (Sievers NOA 280i; Boulder, CO, USA). Blood samples were collected in EDTA-containing tubes and then centrifuged at 2000 ×g for 15 minutes at 4°C. The plasma was frozen and stored at –80°C until analysis. Plasma TNF-α, MCP-1, IL-1β, and TGF-β levels were measured using the enzyme-linked immunosorbent assay (ELISA) with commercially available kits (Human tumour necrosis factor-alpha (TNF-α) ELISA Kit, cat. no. Ab181421, Abcam; Human monocyte MCP-1 ELISA Kit, cat. no. Ab179886, Abcam; Human IL-1β ELISA Kit, cat. no. Ab46052 Abcam; Human TGF-β ELISA Kit, cat. no. Ab100647 Abcam) according to the manufacturer's instructions. The plasma hs-CRP assay was based on the latex agglutination immunoassay method. When an antigen-antibody reaction occurs between CRP in a sample and anti-CRP antibody, which has been sensitized to latex particles, agglutination results. This agglutination is detected as an absorbance change (572 nm), with the magnitude of the change proportional to the quantity of CRP in the sample. The actual concentration is then determined by interpolation from a calibration curve prepared from calibrators of known concentration. The upper normal value of hs-CRP was 0.3 mg/dl in our laboratory. The plasma levels of ADMA were determined using commercially available enzyme-linked immunosorbent assay (ELISA) kits (DLD Diagnostika, Hamburg, Germany and R&D system). The correlation coefficient between liquid chromatography-mass spectrometry ADMA and ELISA ADMA was 0.98. The recovery rate for ADMA was 95%, and the within-assay and between-assay variation coefficients were not more than 7% and 8%, respectively. The plasma levels of homocysteine were measured using the fluorescence polarization immunoassay based on the highly selective enzymatic conversion of homocysteine to S-adenosyl-L-homocysteine, which was then recognized by a monoclonal antibody (Abbott AxSYM Homocysteine; 5F51-20, IL). The default result unit for AxSYM homocysteine is μmol/L. The plasma levels of MMP-2 and MMP-9 were determined using commercially available ELISA kits according to the manufacturer’s instructions (Amersham Bioscience, Uppsala, Sweden). All the procedures were performed according to the manufacturer’s instructions. Each standard and each plasma sample were analysed twice, and the mean values were used for all subsequent data analyses. 2. How to select the adjusted factors in a multivariable model? In Tables 2 and 4, only age and gender were adjusted in the multivariable analysis. In table 2, access type, PAOD, phosphate, anti-platelet agents, aspirin, and MMP-2 were significant in univariate analysis. In table 4, access type, diuretic, statin, ADMA, and MMP-9 were significant in univariate analysis. In general, we may control all other potential confounders in the multivariable model. Thus, why the author only select age and gender as confounders in this study?Author Reply: Thank you for the comments and suggestions regarding the footnotes of Tables 2 and 4. We have amended the footnotes as follows: multivariate analysis: adjusted variables with significance (P<0.05) in univariate analysis. 3. In Table 4, the ADMA markers in the outcome of arteriovenous fistulas/grafts’ dysfunction were evaluated using dichotomous data but in the outcome of arteriovenous fistulas/grafts’ acute thrombosis using continuous data. Why the analysis process is different. Besides, other markers were all analyzed using continuous data. Thus, AMDA should be treated in the same way as other oxidative stress markers.Author Reply: Thank you for the suggestions and comments for evaluating biomarkers with continuous or dichotomous data in the outcomes of acute thrombosis or dysfunction of arteriovenous fistulas/grafts. In Table 4, we analysed and amended according to your comments with more details on continuous data of all biomarkers in the manuscript as follows: Next, we performed multivariate Cox regression analysis to identify the independent predictors. Only plasma ADMA independently predicted whether progressive dysfunction of access occurred easily in our HD patients (Table 4; hazard ratio (HR): 4.36; P=0.031).
Table 4. Univariate and multivariate Cox regression analysis for the predictors of dysfunction of arteriovenous fistulas/grafts within 12 months of follow-up.
Parameter |
Univariate analysis |
Multivariate analysis |
||
HR (95% Cl) |
P |
HR (95% Cl) |
P |
|
Age (years) |
0.98 (0.95-1.02) |
0.312 |
0.97 (0.94-1.01) |
0.125 |
Sex |
0.506 |
0.872 |
||
Female |
1.00 |
1.00 |
||
Male |
0.76 (0. 34-1.70) |
1.08 (0.44-2.64) |
||
Access type (AVF/AVG) |
||||
AVF |
1.00 |
1.00 |
||
AVG |
2.28 (1.03-5.08) |
0.043* |
2.39 (0.91-6.28) |
0.076 |
Risk factors |
||||
Hypertension (%) |
1.65 (0.49-5.53) |
0.418 |
||
Diabetes mellitus (%) |
1.36 (0.61-3.04) |
0.448 |
||
Current smoking (%) |
0.99 (0.23-4.20) |
0.986 |
||
CAD (%) |
1.26 (0.56-2.83) |
0.581 |
||
PAOD (%) |
0.57 (0.17-1.90) |
0.357 |
||
Plasma biochemical data |
||||
LDL-C (mg/dL) |
1.01 (0.99-1.02) |
0.193 |
||
HDL-C (mg/dL) |
1.00 (0.98-1.03) |
0.794 |
||
TG (mg/dL) |
1.00 (0.99-1.00) |
0.735 |
||
Calcium (mg/dL) |
1.44 (1.00-2.10) |
0.053 |
||
Phosphate (mg/dL) |
1.14 (0.87-1.48) |
0.354 |
||
Albumin (mg/dL) |
1.17 (0.35-3.86) |
0.799 |
||
Creatinine (mg/dL) |
1.09 (0.94-1.26) |
0.258 |
||
Kt/V |
0.71 (0.16-3.13) |
0.652 |
||
Medications |
||||
Anti-platelet agents |
1.35 (0.59-3.07) |
0.482 |
||
Aspirin |
0.96 (0.43-2.14) |
0.918 |
||
Dipyridamole |
1.41 (0.48-4.12) |
0.532 |
||
Cilostazol |
0.77 (0.10-5.67) |
0.793 |
||
Coumadin |
3.06 (0.72-13.03) |
0.13 |
||
Nitrates |
1.10 (0.49-2.47) |
0.821 |
||
β-blockers |
0.89 (0.38-2.07) |
0.777 |
||
Calcium antagonists |
0.81 (0.36-1.80) |
0.603 |
||
ACEI/ARB |
0.99 (0.41-2.38) |
0.973 |
||
Diuretic |
4.23 (1.85-9.66) |
0.001* |
1.80 (0.63-5.16) |
0.277 |
Lipid-lowering agents |
||||
Statin |
2.80 (1.05-7.50) |
0.040* |
1.16 (0.35-3.83) |
0.810 |
Fibrate |
0.05 (<0.0001-3221.32) |
0.592 |
||
Plasma biomarkers |
||||
Hs-CRP (mg/dL) |
1.01 (0.83-1.42) |
0.907 |
||
Homocysteine (μmol/L) |
0.99 (0.95-1.03) |
0.634 |
||
ADMA (μmol/mL) |
13.90 (3.20 - 60.45) |
<0.001* |
4.36 (1.14-16.65) |
0.031*,a |
MMP-2 (ng/mL) |
0.999 (0.997-1.001) |
0.471 |
||
MMP-9 (ng/mL) |
1.01 (1.00-1.02) |
0.011* |
1.00 (0.99-1.01) |
0.602 |
NO (μM) |
1.00 (1.000-1.002) |
0.089 |
||
MCP-1 (pg/mL) |
1.00 (0.997-1.002) |
0.711 |
||
TGF-β (pg/mL) |
1.01 (0.96-1.06) |
0.860 |
||
TNF-α (pg/mL) |
0.99 (0.98-1.00) |
0.204 |
||
IL-1β (pg/mL) |
1.01 (0.999-1.015) |
0.086 |
Footnote: a: The baseline plasma ADMA level of 0.6207 μmol/mL was identified as the best cut-off value for predicting symptomatic dysfunction of access (Figure 3); therefore, ADMA >0.6207 μM was used as an indicator of plasma biomarkers in Supplementary Table 1. In multivariate analysis, the plasma baseline levels of ADMA independently conferred nearly 4.55 times the risk of primary stenotic dysfunction of HD VA (hazard ratio: 4.55; 95% confidence interval: 1.20 to 17.26, P=0.026) (shown in Supplementary Table 1).
*: p-value<0.05; multivariate analysis: adjusted variables with significance (p<0.05) in univariate analysis.
Abbreviations: ACEI, angiotensin-converting enzyme inhibitor; ADMA, asymmetrical dimethylarginine; ARB, angiotensin receptor blocker; AVF, arteriovenous fistula; AVG, arteriovenous graft; CAD, coronary artery disease; HDL-C, high-density lipoprotein cholesterol; Hs-CRP, high-sensitivity C-reactive protein; LDL-C, low-density lipoprotein cholesterol; MMP-2, matrix metalloproteinase-2; MMP-9, matrix metalloproteinase-9; NO, nitrate oxidase; PAOD, peripheral arterial occlusive disease; TG, triglyceride; IL-1β: interleukin-1β; MCP-1: monocyte chemotactic protein 1; TNF-α: tumour necrosis factor-α; TGF-β: transforming growth factor-β. Supplementary Table 1
Table S1. Univariate and multivariate Cox regression analyses for the predictors of dysfunction of arteriovenous fistulas/grafts within 12 months of follow-up.
Parameter |
Univariate analysis |
Multivariate analysis |
||
HR (95% Cl) |
P |
HR (95% Cl) |
P |
|
Age (years) |
0.98 (0.95-1.02) |
0.312 |
0.97 (0.94 - 1.01) |
0.124 |
Sex |
0.506 |
0.862 |
||
Female |
1.00 |
1 |
||
Male |
0.76 (0. 34-1.70) |
1.08 (0.44 - 2.66) |
|
|
Access type (AVF/AVG) |
||||
AVF |
1.00 |
1 |
||
AVG |
2.28 (1.03-5.08) |
0.043* |
2.42 (0.92 - 6.35) |
0.073 |
Risk factors |
||||
Hypertension (%) |
1.65 (0.49-5.53) |
0.418 |
||
Diabetes mellitus (%) |
1.36 (0.61-3.04) |
0.448 |
||
Current smoking (%) |
0.99 (0.23-4.20) |
0.986 |
||
CAD (%) |
1.26 (0.56-2.83) |
0.581 |
||
PAOD (%) |
0.57 (0.17-1.90) |
0.357 |
||
Plasma biochemical data |
||||
LDL-C (mg/dL) |
1.01 (0.99-1.02) |
0.193 |
||
HDL-C (mg/dL) |
1.00 (0.98-1.03) |
0.794 |
||
TG (mg/dL) |
1.00 (0.99-1.00) |
0.735 |
||
Calcium (mg/dL) |
1.44 (1.00-2.10) |
0.053 |
||
Phosphate (mg/dL) |
1.14 (0.87-1.48) |
0.354 |
||
Albumin (mg/dL) |
1.17 (0.35-3.86) |
0.799 |
||
Creatinine (mg/dL) |
1.09 (0.94-1.26) |
0.258 |
||
Kt/V |
0.71 (0.16-3.13) |
0.652 |
||
Medications |
||||
Anti-platelet agents |
1.35 (0.59-3.07) |
0.482 |
||
Aspirin |
0.96 (0.43-2.14) |
0.918 |
||
Dipyridamole |
1.41 (0.48-4.12) |
0.532 |
||
Cilostazol |
0.77 (0.10-5.67) |
0.793 |
||
Coumadin |
3.06 (0.72-13.03) |
0.13 |
||
Nitrates |
1.10 (0.49-2.47) |
0.821 |
||
β-blockers |
0.89 (0.38-2.07) |
0.777 |
||
Calcium antagonists |
0.81 (0.36-1.80) |
0.603 |
||
ACEI/ARB |
0.99 (0.41-2.38) |
0.973 |
||
Diuretic |
4.23 (1.85-9.66) |
0.001* |
1.80 (0.62 - 5.16) |
0.277 |
Lipid-lowering agents |
||||
Statin |
2.80 (1.05-7.50) |
0.040* |
1.16 (0.35 - 3.85) |
0.810 |
Fibrate |
0.05 (<0.0001-3221.32) |
0.592 |
||
Plasma biomarkers |
||||
Hs-CRP (mg/dL) |
1.01 (0.83-1.42) |
0.907 |
||
Homocysteine (μmol/L) |
0.99 (0.95-1.03) |
0.634 |
||
ADMA >0.6207 (μmol/mL) |
7.20 (2.44 - 21.30) |
<0.001* |
4.55 (1.20 - 17.26) |
0.026* |
MMP-2 (ng/mL) |
0.999 (0.997-1.001) |
0.471 |
||
MMP-9 (ng/mL) |
1.01 (1.00-1.02) |
0.011* |
1.00 (0.99 - 1.01) |
0.712 |
NO (μM) |
1.00 (1.000-1.002) |
0.089 |
||
MCP-1 (pg/mL) |
1.00 (0.997-1.002) |
0.711 |
||
TGF-β (pg/mL) |
1.01 (0.96-1.06) |
0.860 |
||
TNF-α (pg/mL) |
0.99 (0.98-1.00) |
0.204 |
||
IL-1β (pg/mL) |
1.01 (0.999-1.015) |
0.086 |
*: p-value<0.05; multivariate analysis: adjusted variables with significance (p<0.05) in univariate analysis.
Abbreviations: AVF, arteriovenous fistula; AVG, arteriovenous graft; Hs-CRP, high-sensitivity C-reactive protein; LDL-C, low-density lipoprotein cholesterol; MMP-2, matrix metalloproteinase-2; MMP-9, matrix metalloproteinase-9; NO, nitrate oxidase; PAOD, peripheral arterial occlusive disease; TG, triglyceride; IL-1β: interleukin-1β; MCP-1: monocyte chemotactic protein 1; TNF-α: tumour necrosis factor-α; TGF-β: transforming growth factor-β. 4. Please demonstrate the ROC curve of ADMA values for the rate of dysfunction-free survival and provide the best cut-off point in the figure.Author Reply: We agree with your comment and have amended the manuscript as follows:
Based on receiver operating characteristic analysis, a baseline plasma ADMA level of 0.6207 μmol/mL was identified as the best cut-off value for predicting symptomatic dysfunction of access (Figure 3). The patients were divided into high (>0.6207 μM) versus low (≤0.6207 μM) ADMA groups (shown in Table 5). In the high ADMA group, 27.7% of patients had VA dysfunction requiring PTA during the follow-up period compared with 4.4% in the low ADMA group (P<0.001).
Figure 3. ROC curve of ADMA values for the rate of dysfunction-free survival and provide the best cut-off point
- Since access type had a significant impact on the outcome (acute thrombosis or AV dysfunction), a subgroup analysis to investigate the oxidative stress markers and outcome association stratifies the vascular access type.Author Reply: Thank you for your invaluable comments and suggestions. We have amended the manuscript as follows: Because access type had a significant impact on the VA outcome, subgroup analysis was performed as shown in Supplementary Table 2 (acute thrombosis) and Supplementary Table 3 (dysfunction). After multivariate analysis, no significance was found between plasma biomarkers and acute thrombosis or dysfunction of VA among the AVG group. Among the AVF group, homocysteine showed significance with acute thrombosis (HR: 1.08; P=0.004); however, ADMA played an important role in predicting progressive stenotic dysfunction (HR: 28.93; P=0.001).
Supplementary Table 2
Table S2. Oxidative stress markers and acute thrombosis stratified by vascular access type
|
AVF |
AVG |
||||
|
Acute thrombosis (n=3) |
Non-thrombosis (n=103) |
p-value |
Acute thrombosis (n=21) |
Non-thrombosis (n=32) |
p-value |
hsCRP (mg/dL) |
0.18±0.18 |
1.11±2.19 |
0.355 |
1.05±1.19 |
1.28±1.92 |
0.723 |
Homocysteine (μmol/L) |
56.02±27.21 |
27.65±11.35 |
0.008* |
26.27±8.29 |
24.56±6.29 |
0.454 |
ADMA (μmol/mL) |
0.85±0.13 |
0.57±0.26 |
0.05 |
0.65±0.28 |
0.62±0.24 |
0.707 |
MMP2 (ng/mL) |
685.00±239.62 |
866.02±211.64 |
0.233 |
771.04±175.64 |
820.30±199.51 |
0.304 |
MMP9 (ng/mL) |
94.80±40.89 |
59.36±39.99 |
0.118 |
57.45±40.45 |
72.51±43.22 |
0.118 |
NO (μM) |
303.37±335.87 |
241.78±221.86 |
0.992 |
207.40±178.34 |
306.97±471.86 |
0.422 |
MCP-1 (pg/mL) |
454.19±75.12 |
334.46±183.16 |
0.058 |
311.73±125.45 |
345.39±162.36 |
0.366 |
TGF-β (pg/mL) |
7.70±6.86 |
9.20±9.43 |
0.902 |
4.19±4.36 |
4.91±4.12 |
0.348 |
TNF-α (pg/mL) |
23.48±NA |
76.78±106.09 |
0.788 |
69.99±118.89 |
40.00±70.95 |
0.494 |
IL-1β (pg/mL) |
102.26±48.68 |
51.33±38.06 |
0.068 |
67.48±38.71 |
73.81±45.99 |
0.809 |
Acute thrombosis of AVF
Independent variable |
univariate analysis HR (95% CI) |
p-value |
multivariate analysis HR (95% CI) |
p-value |
Homocysteine |
1.08 (1.03 - 1.14) |
0.002* |
1.08 (1.03 - 1.15) |
0.004* |
ADMA |
22.54 (0.82 - 618.32) |
0.065 |
39.81 (0.46 - 3441.08) |
0.105 |
*: p-value<0.05; multivariate analysis: adjusted variables with significance (p<0.05) in univariate analysis.
Abbreviations: AVF, arteriovenous fistula; AVG, arteriovenous graft; Hs-CRP, high-sensitivity C-reactive protein; LDL-C, low-density lipoprotein cholesterol; MMP-2, matrix metalloproteinase-2; MMP-9, matrix metalloproteinase-9; NO, nitrate oxidase; PAOD, peripheral arterial occlusive disease; TG, triglyceride; IL-1β: interleukin-1β; MCP-1: monocyte chemotactic protein 1; TNF-α: tumour necrosis factor-α; TGF-β: transforming growth factor-β.
Supplementary Table 3
Table S3. Oxidative stress markers and dysfunction stratified by vascular access type
|
AVF |
AVG |
||||
|
Dysfunction (n=12) |
Non-Dysfunction (n=94) |
p-value |
Dysfunction (n=12) |
Non-Dysfunction (n=41) |
p-value |
hsCRP (mg/dL) |
1.15±2.26 |
1.08±2.16 |
0.333 |
1.03±1.16 |
1.24±1.79 |
0.823 |
Homocysteine (μmol/L) |
28.20±6.14 |
28.49±13.34 |
0.414 |
24.34±6.60 |
25.49±7.32 |
0.637 |
ADMA (μmol/mL) |
0.82±0.33 |
0.55±0.23 |
0.007* |
0.73±0.20 |
0.60±0.26 |
0.111 |
MMP2 (ng/mL) |
845.29±237.34 |
862.67±211.70 |
0.846 |
746.28±203.67 |
816.73±185.70 |
0.372 |
MMP9 (ng/mL) |
82.00±43.01 |
57.83±39.38 |
0.064 |
81.75±52.60 |
62.09±38.57 |
0.21 |
NO (μM) |
252.67±123.99 |
242.35±233.88 |
0.202 |
463.00±719.22 |
210.30±185.82 |
0.13 |
MCP-1 (pg/mL) |
340.27±108.55 |
337.81±189.44 |
0.429 |
323.05±99.99 |
333.83±160.12 |
0.594 |
TGF-β (pg/mL) |
11.33±8.35 |
8.87±9.44 |
0.21 |
5.16±3.06 |
4.45±4.54 |
0.283 |
TNF-α (pg/mL) |
21.28±16.66 |
82.16±109.53 |
0.153 |
6.27±3.21 |
63.24±95.50 |
0.053 |
IL-1β (pg/mL) |
70.04±35.51 |
50.72±39.20 |
0.059 |
80.85±44.74 |
68.00±42.24 |
0.421 |
Dysfunction of AVF
Independent variable |
HR (95% CI) |
p-value |
ADMA |
28.93 (3.94 - 212.58) |
0.001* |
*: p-value<0.05; multivariate analysis: adjusted variables with significance (p<0.05) in univariate analysis.
Abbreviations: AVF, arteriovenous fistula; AVG, arteriovenous graft; Hs-CRP, high-sensitivity C-reactive protein; LDL-C, low-density lipoprotein cholesterol; MMP-2, matrix metalloproteinase-2; MMP-9, matrix metalloproteinase-9; NO, nitrate oxidase; PAOD, peripheral arterial occlusive disease; TG, triglyceride; IL-1β: interleukin-1β; MCP-1: monocyte chemotactic protein 1; TNF-α: tumour necrosis factor-α; TGF-β: transforming growth factor-β.
6. As for the ADMA biomarkers, the comparison between the UPLC-MS/MS method and the ELISA showed only a moderate correlation. Therefore, this needs to be taken into account when considering absolute concentrations. A reference could be cited and discussed in the study limitation (Jente Boelaert et al. Toxins. 2016 May 13;8(5):149).Author Reply: Thank you for the suggestion. We have amended the manuscript, including the limitation and reference as follows: Regarding the ADMA biomarkers, the comparison between the UPLC-MS/MS method and ELISA showed only a moderate correlation. Therefore, this finding must be considered with absolute concentrations (Jente Boelaert et al. Toxins. 2016 May 13;8(5):149). 7. The author demonstrated table 5 to illustrate the patients' characteristics differences between high and low ADMA levels. However, a correlation matrix between clinical factors and oxidative stress markers would also be interesting to find the possible clinical effect on ADMA level in hemodialysis patients.Author Reply: Thank you for the suggestion. We have amended the manuscript and Supplementary Table 4 as follows: Furthermore, the correlation matrix between clinical factors and oxidative stress markers is shown in Supplementary Table 4, which also reveals the high correlation between ADMA and the dysfunction of AVF. Supplementary Table 4 Table S4. Correlation matrix between clinical factors and oxidative stress markers (Pearson’s correlation statistics) 8. A higher level of ADMA was presented a higher VA dysfunction rate. Patients with a higher level of ADMA presented fewer duration of access use, more diuretics used, a higher level of LDL, calcium, homocysteine, MMP-9, and IL-1 beta, as well as a lower level of TNF-alpha. A brief discussion was recommended to explain these findings.Author Reply: Thank you for the suggestion. We have amended the manuscript and added reference [28] as follows: Discussion IL-1β, another major factor specific to AVF, can also induce inflammation not only because of AVF creation but also because of repeat needle stick injury. Platelets can adhere to the already inflamed endothelial tissue and potentiate the process by releasing IL-1β, MCP-1, TNF-α, and other inflammatory cytokines. These cytokines can further cause the activation cascade leading to increased inflammation, adhesion, and eventually plaque or thrombus formation [25–27]. Our study showed that a higher level of ADMA is associated with a higher level of MMP-9 and IL-1β, indicating inflammation and endothelial dysfunction can promote stenosis dysfunction of VA and lead to shorter duration of VA use. The ADMA level is significantly related to higher levels of LDL and homocysteine, suggesting the clear relationship to atherosclerosis and cardiovascular events, which could drive further progression of CKD and consequently more diuretics used. ADMA can regulate endothelial NO production by modulating the calcium-sensing receptor, thus increasing intracellular calcium release. This finding may also explain the higher level of serum calcium among patients with a higher ADMA level in this study [28]. TNF-α, a 17-kD protein, is a prominent inflammatory cytokine of interest in intimal hyperplasia. Unexpectedly, our study showed a lower level of TNF-α in patients with a higher level of ADMA. TNF-α has apoptotic properties, which can be masked by the antiapoptotic NF-kB pathway that it activates. Consequently, only when the NF-kB pathway and/or protein synthesis are nonfunctional does TNF-α become apoptotic. This finding may also explain why TNF-α usually appears to be proliferative with intimal hyperplasia [16]. Although many cells may show cellular injury at the time of AVF placement, it is insufficient to compromise the antiapoptotic effects of the NF-kB pathway. Thus, TNF-α in the setting of intimal hyperplasia can contribute synergistically with the proliferative effects of other cytokines released during AVF placement [25–27].
Finally, the English language of the revised manuscript was edited by AJE. We sincerely appreciate your time and effort spent in reviewing this manuscript. In reviewing and revising our manuscript, we are motivated to read more and, thus, learn more from your criticisms.

Reviewer 2 Report
Thank you very much for allowing me to review the original article "The Role of Oxidative Stress Markers in the Prediction of Acute Thrombotic Occlusion of Haemodialysis Vascular Access and Progressive Stenotic Dysfunction Demanding Angioplasty" (antioxidants-1149758).
The problem that this work addresses is that we know that vascular dialysis (VA) dysfunction can increase morbidity and mortality in haemodialysis (HD) patients. This work arises from the hypothesis of if the oxidative stress markers can be used as predictors for thrombotic occlusion of VA and progressive stenosis dysfunction demanding percutaneous transluminal angioplasty (PTA).
This is a pilot study in one teaching hospital. A total of 165 HD patients were recruited of then 159 patients (83 males, 76 females, 30mean age: 65 ± 12) were followed up clinically for up to 12 months to estimate the amount of primary thrombotic occlusion and VA dysfunction demanding PTA.
Their results suggest the role of ADMA in the development of symptomatic VA dysfunction. Additionally, PAOD severity can be used in clinical practice to predict whether acute thrombotic occlusion of VA will easily occur in HD patients.
Comments:
Please explain the following aspects:
Follow-up 12 months… .control 6 months… 2 controls?
Line 130-132, indicate the total number of patients excluded and for each reason.
Line 186-187, indicate the monitoring procedure and its periodicity during the year that the study lasted.
Line 210, indicate the adjustment variables used in the multivariate model. Figure 1 does not correspond to results, it must be included in the methods section.
Line 218, the cumulative incidence of the acute venous thrombosis of the study should be indicated.
Line 287, indicate the mean period of follow-up of the cumulative delivery of the acute venous thrombosis in the study.
Figure 2 should be commented.
Line 381, eliminate "we demonstrated that" as it is a pilot study with a small sample size, the results must be adjusted.
Figure 3 should be in results.
This is a pilot study with interesting results, but it has a small sample size and this cohort was cross-sectional in design to estimate the primary assisted patency not from the initiation of each VA construction, these aspects should be taken into account in the conclusions.
Author Response
Reviewer 2
Thank you very much for allowing me to review the original article "The Role of Oxidative Stress Markers in the Prediction of Acute Thrombotic Occlusion of Haemodialysis Vascular Access and Progressive Stenotic Dysfunction Demanding Angioplasty" (antioxidants-1149758). The problem that this work addresses is that we know that vascular dialysis (VA) dysfunction can increase morbidity and mortality in haemodialysis (HD) patients. This work arises from the hypothesis of if the oxidative stress markers can be used as predictors for thrombotic occlusion of VA and progressive stenosis dysfunction demanding percutaneous transluminal angioplasty (PTA). This is a pilot study in one teaching hospital. A total of 165 HD patients were recruited of then 159 patients (83 males, 76 females, 30mean age: 65 ± 12) were followed up clinically for up to 12 months to estimate the amount of primary thrombotic occlusion and VA dysfunction demanding PTA. Their results suggest the role of ADMA in the development of symptomatic VA dysfunction. Additionally, PAOD severity can be used in clinical practice to predict whether acute thrombotic occlusion of VA will easily occur in HD patients.
Author Reply: We sincerely appreciate your time and effort spent in reviewing this manuscript. We have revised the manuscript thoroughly according to your suggestions. The responses to your comments are found below.
Comments:
Please explain the following aspects:
1. Follow-up 12 months… .control 6 months… 2 controls?Author Reply: Thank you for the suggestion. We have amended the Abstract, Materials and Methods (Study Participants) and Figure 1 as follows: The participants met the following criteria: (1) received regular HD treatment for at least 6 months, without clinical evidence of acute or chronic inflammation, recent myocardial infarction, unstable angina or circulatory congestion; and (2) received an arteriovenous fistula (AVF)/arteriovenous graft (AVG: polytetrafluoroethylene, PTFE) as the current VA for more than 6 months, without interventions within the last 6 months. All the participants were followed up clinically for up to 12 months to estimate the amount of primary thrombotic occlusion and VA dysfunction demanding PTA.
Footnote: The participants met the following criteria: (1) received regular dialysis treatment for at least 6 months, without clinical evidence of acute or chronic inflammation, recent myocardial infarction, unstable angina or circulatory congestion; and (2) received an arteriovenous fistula (AVF)/arteriovenous graft (AVG: polytetrafluoroethylene, PTFE) as the current VA for more than 6 months, without interventions within the last 6 months. All the participants were followed up clinically for up to 12 months to estimate the amount of primary thrombotic occlusion and VA dysfunction demanding PTA.
Figure 1. Flow chart of the study participants in prospective cohort trials to evaluate biomarkers and survival either between thrombotic and non-thrombotic haemodialysis (HD) patients or between progressive dysfunction demanding PTA and intact maintenance function of VA
2. Line 130-132, indicate the total number of patients excluded and for each reason.Author Reply: Thank you for the suggestion. We have amended the Results section as follows:
Among them, 6 patients were excluded: two because of the use of a Hickman catheter with immature VA and four because of receiving the VA intervention procedure within the last 6 months of the start of this project. Therefore, the study group comprised 159 patients (83 men and 76 women; mean age: 65 ± 12 years).
3. Line 186-187, indicate the monitoring procedure and its periodicity during the year that the study lasted.
Author Reply: Thank you for the suggestion. We have amended the supplementary manuscript as follows: According to the KDOQI Work Group recommendation on Guideline Statements 15.1 [23], local resources and the severity of findings on clinical monitoring should determine the timeframe, choice, and extent of imaging studies for further evaluation of which timeframe of less than 2 weeks was deemed reasonable periodicity. In particular, the clinical indicators (signs and symptoms) were suggested to follow up clinical and significant lesions during access monitoring procedures including 2, 1. Physical examination or check such as
1.1. Ipsilateral extremity edema
1.2. Alterations in the pulse, with a weak or resistant pulse, difficult to compress, in the area of stenosis
1.3. Abnormal thrill (weak and/or discontinuous) with only a systolic component in the region of stenosis
1.4. Abnormal bruit (high pitched with a systolic component in the area of stenosis)
1.5. Failure of the fistula to collapse when the arm is elevated (outflow stenosis) and lack of pulse augmentation (inflow stenosis)
1.6. Excessive collapse of the venous segment upon arm elevation 2. Dialysis such as
2.1. New difficulty with cannulation when previously not a problem
2.2. Aspiration of clots
2.3. Inability to achieve the target dialysis blood flow
2.4. Prolonged bleeding beyond usual for that patient from the needle puncture sites for 3 consecutive dialysis sessions
2.5. Unexplained (>0.2 units) decrease in the delivered dialysis dose (Kt/V) on a constant dialysis prescription without prolongation of dialysis duration
4. Line 210, indicate the adjustment variables used in the multivariate model. Figure 1 does not correspond to results, it must be included in the methods section.Author Reply: We agree with your comment and have amended the manuscript and moved Figure 1 to the Methods section as follows: Adjusted variables with P<0.05 by univariate analysis were included in the multivariate model. Methods
2.1. Study Participants
One hundred sixty-five patients who were treated with HD were recruited to this study, and their data were collected at Armed Forces Taoyuan General Hospital, Taiwan, from February 2019 to January 2020 (Figure 1).
5. Line 218, the cumulative incidence of the acute venous thrombosis of the study should be indicated.Author Reply: Thank you for the comments. We have amended the Results section in the manuscript as follows: The cumulative incidence of acute venous thrombosis in the study was 24 (24/159= 15.09%).
6. Line 287, indicate the mean period of follow-up of the cumulative delivery of the acute venous thrombosis in the study.Author Reply: Thank you for the suggestion. We have amended the manuscript to include the mean period of follow-up for the cumulative delivery of both acute thrombotic occlusion and progressive dysfunction. Results section in the manuscript as follows
3.3. Baseline Parameters and Acute Thrombotic Occlusion
The characteristics of the 135 non-thrombotic VA subjects and 24 subjects with sudden-onset thrombotic occlusion of VA are listed in Table 1. The mean period of follow-up for the cumulative delivery of acute thrombotic occlusion was 3.67 ± 2.66 months.……….
3.4. Baseline Parameters and Progressive Dysfunction of Vascular Access
Twenty-four patients had dysfunction of dialysis VA (which fulfilled the 2006 DOQI clinical practice guidelines for VA) and required referral for evaluation and treatment (PTA) [21]. During this one-year prospective follow-up period, the cumulative incidence of the dysfunction was 15.09% (24/159). 50% (12/24) of the patients experienced restenosis after the PTA procedure at the same location. The other 12 patients with the subsequent well function of VA did not receive PTA again. The mean period of follow-up for the cumulative delivery of progressive dysfunction was 4.18 ± 3.08 months. However, 4 non-restenosis patients were lost to follow-up owing to death from infectious or cardiac diseases.
7. Figure 4 should be commented.Author Reply: Thank you for the suggestion. We have amended the Results section in the manuscript as follows: Kaplan-Meier analysis showed that high (>0.6207 μM) ADMA was associated with significantly lower sustained access function (p< 0.001)—i.e., dysfunction with higher ADMA (Figure 4).
8. Line 381, eliminate "we demonstrated that" as it is a pilot study with a small sample size, the results must be adjusted.Author Reply: Thank you for the suggestion. We have amended the manuscript as follows: The present investigation was a pilot study with a small sample size in addition to the VA type used, and PAOD comorbidity is another strong predictor in acute thrombotic VA. PAOD….
9. Figure 2 should be in results.Author Reply: Thank you for the suggestion. We have amended the manuscript as follows: Results… Next, we performed multivariate Cox regression analysis to identify the independent predictors. Only access type of AVG (HR: 16.93, P<0.001) and history of PAOD (HR: 2.35, P=0.047) were independent predictors of acute thrombosis risk in our HD patients (Table 2). Further analysis after combining these two factors—namely, VA graft type and PAOD—revealed an acute thrombotic event rate of up to 50%-60% in our patients during the one-year follow-up (Figure 2). DiscussionIn our study, the risk of acute thrombosis occlusion in AVG with PAOD was higher, significantly approaching 60%.
10. This is a pilot study with interesting results, but it has a small sample size and this cohort was cross-sectional in design to estimate the primary assisted patency not from the initiation of each VA construction, these aspects should be taken into account in the conclusions.Author Reply: Thank you for the suggestion. We have amended the Limitations and Conclusions section as follows: LimitationsThird, our cohort was cross-sectional in design to estimate the primary assisted patency not from the initiation of each VA construction. There could be some intervention therapies before enrolment in our study. ConclusionsOur pilot study presented encouraging results but had a small sample size and this cohort was cross-sectional in design, estimating the primary assisted patency not from the initiation of each VA construction of HD patients to understand which factors was statistical significance in HD VA…
Finally, the English language of the revised manuscript was edited by AJE. We sincerely appreciate your time and effort spent in reviewing this manuscript. In reviewing and revising our manuscript, we are motivated to read more and, thus, learn more from your criticisms.

Round 2
Reviewer 1 Report
All comments have been addressed adequately. I have no further suggestion.
Reviewer 2 Report
After reviewing the manuscript and the comments of the reviewers of the original article "The Role of Oxidative Stress Markers in the Prediction of Acute Thrombotic Occlusion of Haemodialysis Vascular Access and Progressive Stenotic Dysfunction Demanding Angioplasty" (antioxidants-1149758).
I believe that the authors have improved the understanding of their work for the readers of this article.
I consider this to be a pilot study in which the results are preliminary, so I encourage these authors to continue working on this issue.